# A Systematic Literature Review on Privacy by Design in the Healthcare Sector

**Farida Habib Semantha, Sami Azam, Kheng Cher Yeo and Bharanidharan Shanmugam ***

College of Engineering, IT and Environment, Charles Darwin University, Casuarina 0810, Australia;
faridahabib.semantha@cdu.edu.au (F.H.S.); sami.azam@cdu.edu.au (S.A.); Charles.Yeo@cdu.edu.au (K.C.Y.)

**\*** Correspondence: bharanidharan.shanmugam@cdu.edu.au (B.S.)

**Abstract:** In this digital age, we are observing an exponential proliferation of sophisticated hardware- and software-based solutions that are able to interact with the users at almost every sensitive aspect of our lives, collecting and analysing a range of data about us. These data, or the derived information out of it, are often too personal to fall into unwanted hands, and thus users are almost always wary of the privacy of such private data that are being continuously collected through these digital mediums. To further complicate the issue, the infringement cases of such databanks are on a sharp rise. Several frameworks have been devised in various parts of the globe to safeguard the issue of data privacy; in parallel, constant research is also being conducted on closing the loopholes within these frameworks. This study aimed to analyse the contemporary privacy by design frameworks to identify the key limitations. Seven contemporary privacy by design frameworks were examined in-depth in this research that was based on a systematic literature review. The result, targeted at the healthcare sector, is expected to produce a high degree of fortification against data breaches in the personal information domain.

**Keywords:** privacy by design; data protection; data breach; healthcare sector; frameworks

## 1. Introduction

Software systems dealing with sensitive and personal user data are facing critical roadblocks regarding the issue of maintaining high level of effective data privacy in recent times [1]. In any information system, privacy and confidentiality are the basic security goals to be taken into consideration. Most e-services depend on stored data to identify a user's personal and medical records. [2]. In spite of its importance, the issue of privacy is oftentimes taken as an afterthought, mainly because of insufficient expertise/awareness among system designers and developers [1]. Security requirements may bypass the foundational goals of privacy if privacy requirements are not implemented with enough clarity [2].

Contemporary challenges of data protection and implementing the right regulatory standards to the organisation's assets against the growing threat of cyber threats are a big concern to the organisations around the world. The 2018 U.S. Department of Health and Human Services (HHS) report states that 6.1 million individuals are affected by 229 healthcare data breaches, and this information was reported to civil rights of breach portal [3]. Statista mentions about the number of data breaches in the U.S. are on an upward trend since 2005. A total of 783 data breaches were reported in 2014, with around 85.61 million total records having been compromised, which was an increase of approximately 500% from 2005 [4]. That number increased more than double to 1579 in 3 years reported in 2017 [4]. Likewise, between 1 April to 30 June of 2018, healthcare service providers suffered the highest number of data breaches than any other sector in Australia, where 49 out of 242 breaches were reported by healthcare sectors because of human errors as per the Office of the Australian Information Commissioner (OAIC) [5]. The average financial loss of a breach involving 1 million records is nearly AUD 40 million [3]. Currently

healthcare sectors in Australia have been automated, wherein patients' personal records, clinical data, warehousing, and other related information is stored via electronic medium [6]. Patients, medical practitioners, and medical service providers are collaborating with more sensitive information than ever before. Recent major threats to data privacy occurred due to unauthorised access, data theft, data loss, and improper data disposal and IT hacking incidents [7]. Healthcare organizations had the highest costs associated with single data breach at AUD 408, which is three times higher than average [3]. Many of them have even started to adopt various Blockchain-based solutions to implement secure decentralized Blockchain-based patient records and monitoring patient progress, both remote and onsite, with full data protection against leaking to a third party [8].

Privacy by design (PbD) is a process based on proactively embedding good privacy practices into the design and operation of IT systems, physical infrastructure, and business practices [9]. PbD is aimed at ensuring privacy and gaining personal control over individuals' information, resulting in a sustainable competitive advantage for organizations [10]. The risk of data breaches increases each year, with many organisations being victims of data breaches around the world, having so far struggled to find effective solutions [11]. Nowadays, to make the user trust the system to perform everyday activities for their personal or professional needs is a major challenge in the field of software engineering.

To settle this concern, experts working with data protection mechanisms have advocated in support of the practice of privacy by design strategies that takes into account the privacy requirements beginning straight from the foundation of the design phase. Privacy is identified by the PbD concept as a design criterion that is critically required as being considered during the design stage of the system [12].

The inspiration of this research initiative was to dissect the gaps that have been identified over time in the study of data privacy. Solutions available at current times are mostly falling behind the constant innovativeness of cybercriminals. This fact alone is enough to justify the emergence of privacy by design. This work critically evaluates the recent solutions by applying a systematic literature review to identify the options for further improvement. This paper also deliberates evaluation of existing privacy by design frameworks to highlight the limitations and identifies an exclusive detailed critical understanding on the basis of the reviews. A brief contribution to this literature is as follows: (1) presenting systematic literature review (SLR) method privacy by design framework in the healthcare sector, (2) addressing a discussion of the main challenges, and (3) providing future research directions.

### 1.1. Research Problem

In the current Information and Communication Technology (ICT) environment, particularly for internet users, privacy invasion is considered a major issue. Despite more than 30 years of history, and countless methodologies, advice, and books, the level of infringement of personal data vaults has reached to an all-time high. There is a number of privacy by design frameworks that are proposed and tested, however, the recent solutions are lagging behind in minimizing the rate of data breaches. Due to continuous failure, healthcare providers are beyond delivering expected value and privacy by design frameworks perform as a vital role.

### 1.2. Aim

The aim of this research was to critically analyse and identify the key limitations of existing privacy by design frameworks targeted at the healthcare sector. We applied the systematic literature review (SLR) method to identify the list of publications, and a comparative analysis was conducted among the existing frameworks after filtering on the basis of the criteria. The study also aimed to identify the lacking in these frameworks and propose a viable future research and development direction.

### 1.3. Research Methodology

A systematic literature review (SLR) was used to identify the current gaps existing in data privacy with a scope of healthcare sectors. In this systematic literature review, existing research based on data

breaches and privacy by design that are closely related to healthcare sector were acknowledged. The primary papers selected are closely related to either data privacy or how to mitigate data breaches for both private and public organizations [13]. Both primary and secondary papers were reviewed to obtain further knowledge in this area, which are counted and mentioned in Table 1.

**Table 1.** Sources identified for related research papers.

| Source | Search String | Resource Type | Results | Inclusion Filter | Filter Excluded | Downloaded | Primary | Secondary |
|---|---|---|---|---|---|---|---|---|
| **CDU Library** | 'Data privacy' 'Data breach' 'Privacy by design' 'Healthcare sector' | Articles (779), conference (5) | 784 | Added results outside CDU library | 105 | 65 | 27 | 38 |
| **IEEE Xplore—digital library** | 'Data privacy' 'Data breach' 'Privacy by design' | Conferences (46), courses (12), journals (07), magazines (03), articles (01) | 69 | | 45 | 35 | 23 | 14 |
| **Science Direct** | 'Data privacy' 'Data breach' 'Privacy by design' 'Healthcare sector' | Review articles (78), research articles (371), encyclopaedia (6), book chapters (183), conference abstracts (6), conference info (1), correspondence (1), discussion (16), editorials (6), mini reviews (6), news (15), practice guidelines (1), product reviews (1), short communications (10), other (34) | 735 | Review articles, Research articles Book chapter, Conference abstracts | 77 | 53 | 19 | 34 |
| **CDU Library** | 'Data privacy' 'Data breach' 'Privacy by design' 'Healthcare sector' 'framework' | Article (585), conference (4) | 589 | | 75 | 55 | 22 | 33 |
| **IEEE Xplore—digital library** | 'Data privacy' 'Data breach' 'Privacy by design' 'framework' | Conferences (12), courses (5), journals (2) | 20 | | 10 | 10 | 5 | 6 |
| **Science Direct** | 'Data privacy' 'Data breach' 'Privacy by design' 'Healthcare sector' 'framework' | Review articles (67), research articles (280), encyclopaedia (6), book chapters (122), conference abstracts (4), conference info (1), discussion (9), editorials (2), mini reviews (5), news (10), practice guidelines (1), short communications (6), other (27) | 540 | | 79 | 35 | 12 | 23 |
| **Total** | | | 2780 | | 391 | 253 | 108 | 147 |

Firstly, systematic queries were conducted on a search string of terms such as 'Data privacy', 'Data breach', 'Privacy by design', and 'Healthcare sector' to identify them using three leading databases: Charles Darwin University library search engine, IEEE Xplore digital library, and ScienceDirect. By using this search string, Charles Darwin University library churned out a total of 784 resource types (articles (779) and conference (5)) after inclusion of filters (added results outside CDU library the result of exclusion), with a result of 105. Among them, 65 resources were downloaded. From the downloaded papers, 27 were primary resources and 38 were secondary resources. Accordingly, IEEE Xplore digital library showed a total of 69 resources types (conferences (46), courses (12), journals (07), magazines (03), and articles (01)), and 35 resources among them were downloaded. From the downloaded resources, 22 were primary resources and 13 were secondary resources. Consequently, 22 primary resources were identified via ScienceDirect. Secondly, one additional search string "framework" was added to find out the more relevant results. CDU library showed a total of 589 search results, where 22 resources were identified that were primarily related. IEEE Xplore digital library showed a total of 19 search results,

where four resources were primarily related. Similarly, ScienceDirect showed a total of 540 search results, where 12 resources were primarily related. Charles Darwin University has 252 databases where resources can be searched. This study was based on 132 search results (Table 1). All papers were then refined by applying filters such as search string, resource type, publication date, and English as language. Finally, 253 papers were downloaded on the basis of the selected keywords, where 106 were considered as primary and the remaining 147 were identified as secondary papers [13].

This paper employed a systematic literature review to collect all experimental evidence to review them critically. The objective of this research methodology was not only to collect experimental evidence but also to support the development of guidelines that can then be used by professionals. In our research, the SLR ensured the search and retrieval process impartially and more accurately. A huge number of researches are produced each year, sometimes with conflicting findings. In such a situation, it is difficult to identify which results are most reliable to use. An SLR addresses this problem by identifying, critically evaluating, and integrating the findings of high quality and relevant studies. In our research, this methodology helped to identify the extent of progression of existing research in order to clarify the research problem. This methodology supported us in detecting the relationships, gaps, and inconsistencies in the privacy by design literature, as well as in describing directions for future research. The primary papers selected by this research were closely linked to privacy by design. The contribution of this study was in providing a review of current privacy by design frameworks in order to identify the key limitations. The associated articles that were reviewed are tabled and ordered chronologically.

### 1.4. Relevant Studies

The contributions of our study were in providing a review of current privacy by design in the healthcare system. Most of the primary papers selected for this study conducted their own literature review that was specific to their study. A critical analysis based on some of the primary papers was mentioned. The entire lifecycle of healthcare records needs to be considered while protecting them. This implies that medical data must be accessed only by authorised parties and must not disclose any sensitive data in their distribution [14]. To meet these objectives, technological solutions such as architectures, software tools, privacy models, algorithms, languages, and protocols have been proposed on the basis of the different types of data breaches [15,16]. Moreover, privacy protection is a legal requirement that is posed by the "Privacy Rule of the Health Insurance Portability and Accountability Act (HIPAA)". Significant work has already been completed in this field and, due to the importance of the topic, new privacy by design approaches are continuously emerging. M. Davis, Dr. U Lang, and S. Shetye carried out the work "A cyber model for privacy by design (PbD) based on privacy by design principles" [17]. This model is an executable foundation designed to build interoperable secure privacy capabilities into any standard IT network environment. However, this work is not effective where a poorly secured device is a huge financial risk, particularly in the public domain [18].

Geoff et al. mentioned that it is necessary to construct a common framework in which the approaches can be analysed by putting the previous approaches in perspective [19]. The design of a privacy-sensitive system has to account for the observer and the observed, as well as the connection between them [19]. This framework is designed for implementing privacy measures in ubiquitous computing environments and has demonstrated its application in pervasive healthcare [19]. The gap in this framework is sensitivity of the healthcare environment and its related data, which will play a large part in the adoption applications for pervasive healthcare [20].

Kenthapadi K has researched the topic of "Query auditing model for data privacy" [21]. In this framework, a privacy mechanism is used to query the database that denies the query and alters the answer in order to ensure privacy [22]. A major issue of this research is that query denial may leak information, and thus an attacker can use previously suggested auditors to compromise the privacy of a large fraction of personal data [23].

M. Lieshout, L. Kool, B. Schoonhoven, and M. Jonge's work, "Privacy by design: an alternative to existing practice in safeguarding privacy" introduced PbD to capture different dimensions such as "user perspective", "technical aspects", and organizational and design stage from foundation to a serious perspective by referring to information systems. This work is still in progress, and the framework requires further elaboration and validation [18,24].

V. Ragunatha and S. Manmeet carried out the work, "Privacy-by-design (PbD) IoT framework: A case of location privacy mitigation strategies for near field communication (NFC) tag sensor". This framework offers a privacy solution to secure user filtering or validation with encrypted message, preventing the possibility of retrieving personal information. However, content protection and filtering techniques do not work effectively for proposed framework and have been considered for future enhancement work [25].

The Internet of Things (IoT) plays a major role in several healthcare devices, wherein these devices may be interconnected with wired or wireless networks without user intervention and can transmit a range of private data to and from sources. The IoT has enabled objects to be communicated and information to be exchanged in order to facilitate the collection of advanced intelligent services for users [26]. Chaudhuri and Cavoukian worked on "The proactive and preventive privacy (3P) framework for IoT privacy by design". The Internet of Things (IoT) is vital today in order to ensure that we are not behind in safeguarding data privacy for the multifaceted applications of the emerging IoT [27]. However, this work is not effective in every application and requires additional analysis [28].

Skinner et al. researched the privacy shield framework in organizational information systems. Many privacy enthusiasts and a large section of common mass have feared novel techniques used to effectively confront violent occurrences that have a solemn effect on individuals' rights and ability to protect their personal data privacy. The application of privacy principles in design and the privacy separation of data, public key infrastructure use, and information system policies, constitute a framework to protect users' personal data [29]. This consists of application, anonymity, and concept of separation of data, Hippocratic policy, data repositories, and implementation of public key infrastructure technologies [30,31]. This work is still in progress, and the concept requires validation and compatibility within the healthcare environment [29].

Figure 1 displays various sectors where privacy by design could be applied. Both the SLR and mind map support harmonizing privacy by design, data breach, healthcare sector, and related areas [13].

This mind map develops a central topic "privacy by design" in a radiant structure by organising relevant ideas in a pictured and innovative map. Figure 1 classifies what the top data breach causes are in the healthcare sector, as well as fundamental principles of PbD; PbD strategies; different frameworks, standards, and components of security incident management; core elements of privacy impact assessment; and the lifecycle of data. These are the major components that are identified by SLR to construct the literature review and the related breaches that are connected to these major topics. Indeed, all major topics are connected to the central topic "privacy by design". However, this map supports constructing the overall literature review for this research.

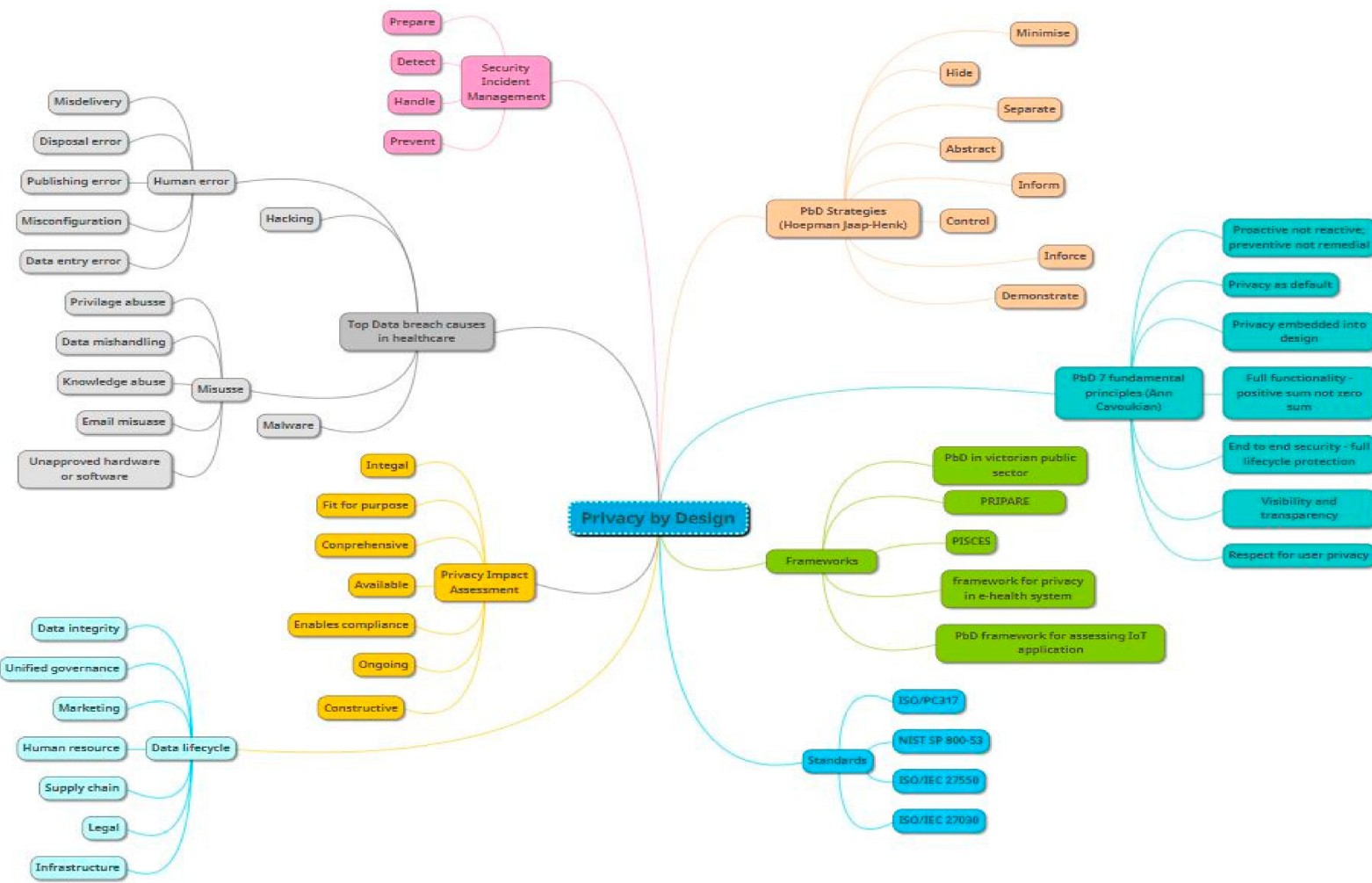

**Figure 1.** Privacy by design mind map [13].

## 2. Analysis on Privacy by Design and Its Key Context

### 2.1. Privacy and Data Protection

The concept of privacy has been a lingering issue for quite some time now, before the entrance of technological advancement. Humans have always wanted to protect their space, family life, house, and conversations, among other things, from unauthorised access. There has always been the need to clearly define boundaries between what is private and what information could be shared with the public. The exact definition of "privacy" is rather debated and open to multiple interpretations [32]. Privacy refers to several versatile social scopes such as social values, individuals' private life, human dignity, freedom of expression and association, and personal autonomy, as well as balance of powers and correspondence. Social value of privacy can be materialized when society benefits from protecting privacy [33]. Privacy thus goes above an individual's own value. Protecting privacy establishes the power to control and limit access to spatial informational, corporeal, psychological, as well as social establishments that circumvent a person [33].

Since the 1970s, data collection practices have been regulated. The U.S. Department of Health Education and Welfare originated a set of privacy principles in 1973 that resulted in the famous Organisation for Economic Co-operation and Development (OECD) fair information principles (FIP) at the beginning of the 1980s [34]. The FIPs are centred on a fair use of data by an individual. These data can be that which holds a one-to-one relationship to a person such as civil status and birthdate. It can also be related in not such a direct fashion, such as the Internet Protocol (IP) address of a device belonging to a specific person. FIP principles define the process of effective information collection and the ways of effective data sharing. These principles project guidelines on the safe storage of data that should be used to shield quality and reliability of data [34]. The cardinal principles are

- There should be a purpose of data collection process based on a defined specification.
- There are no methods available to achieve the purpose in a less invasive way.
- Excessive collection of data should be avoided and the collection process and volume should complement the purpose.
- A necessary safeguarding mechanism should be put in place (quality of data, security measures).
- Data subject rights should be guaranteed (correct data and right to access) [34].

In the course of continuous research attempts on how to safeguard privacy, a relatively novel concept has emerged—the Canadian privacy commissioner (Ontario) Dr. Ann Cavoukian has staunchly advocated this, which is called the "privacy by design" approach [35].

### 2.2. Privacy By Design (PbD)

Privacy by design (PbD) is a series of steps focusing on maintaining and preserving the highest privacy and utilising the strongest data protection possible by incorporating safeguards across the design and development of various services, processes, or products. PbD takes privacy and data protection considerations into the whole developmental process from the start of the development and throughout the whole lifecycle to all types of sensitive information such as healthcare information [19,36].

This privacy by design (PbD) concept was developed and documented by Dr. Ann Cavoukian in the mid-1990s [37–39]. Afterwards, PbD began to be accredited by data protection professionals and regulatory bodies. In October 2010, PbD was unanimously adopted as an international privacy standard at the International Conference of Data Protection and Privacy Commissioners in Jerusalem [40]. Moreover, PbD is included in the U.S. Commercial Privacy Bill of Rights Act. Likewise, it has now been included in the General Data Protection Regulation (GDPR) of the EU and accepted by data protection commissioners worldwide as a concept that will ensure suitable privacy protection in a world of constantly evolving IT systems with the capacity to collect and process massive amounts of data [41].

### 2.3. The Issue of Data Breach in Healthcare Sector.

Healthcare providers play an important role by managing medical data as well as increasing the quality of patient's healthcare through storage, tracking, compilation, collection, and dissemination of both personal and health-related information [42]. Most of the healthcare sectors are now electronic, and therefore electronic records and digital archives are being stored in healthcare facilities as well as within the confinement of various administrative agencies [43].

Although guaranteeing smooth access and unhindered passage of information among the authorised entities, the exchange of electronic health information has further accentuated the need for protecting patients' health information [44]. Health records are thought to be one of the most confidential and sensitive forms of personal data, with the consequence being that personal data infringement is forcing the healthcare providers to face complicated legal and costly financial drawdowns [42,45].

As per Johnson's investigation, data breaches derive from various different sources such as services relating to ambulatory healthcare; hospitals providing acute care; medical laboratories; insurance carriers; healthcare maintenance organizations; and outsource service providers such as transcription, billing, and collection firms [42]. The consequences of data breaches within these entities are diverse. Misuse and leaking of personal health information can bring about serious reputational and/or financial difficulties such as stigmatization and discrimination, ranging from forfeiture of employment and/or insurance cover [44]. Data breach can lead to medical identity fraud, privacy violation, and identity theft—in particular financial identity theft such as fake health insurance, forged taxation, and drug prescription claims. From an organizational perception, financial damage incurred by data infringements consist of both direct costs such as costs associated with clean-up activities and not-so-direct costs such as loss of revenues from reputational harm [42].

Hence, healthcare information privacy is a major ethical and legal issue. Individuals possess the right to exercise control on each and every matter related to their sensitive information, including their own private health information as per the ethical principle of personal autonomy. This right is interpreted into an expectation from general mass and legal clauses, as well as requirements that healthcare providers will provide ample security to the privacy of patients' healthcare information. It should be noted that acceptance of privacy policies and regulatory compliance are not sufficient indicators of complete protection of the privacy of patients' data [46].

Regardless of the legal and ethical obligation of providers of healthcare to shield the privacy of patients' health records, the previous few years have projected a sharp rise in number of reports on healthcare data infringement occurrences [46]. A number of reasons have contributed to such incidents, including (1) from September 2009, the reporting data breaches becoming mandatory; (2) the swiftness at which the healthcare area can be penetrated; and (3) the affluence of sensitive personal data in a patient's health record that could be accessible to criminals such as one's personal health record, which may disclose personal information (date of birth, name, contact number, address, and social security number), financial and insurance details (bank account details, credit card number, insurance numbers), and health data (medications, diagnosis result, addiction problem, allergies, and treatment types) [42].

### 2.4. Fundamental Principles of Pbd in the Context of Healthcare Sector

The concept behind privacy by design has the impact of many privacy experts and a respectable number of advocates, collectively aiding in the distribution and progress of the concept. The concept has its core from Ann Cavoukian's seven fundamental principles, which are stimulated by fair information practices (FIPs) (Figure 2) [35,47,48]. Below are the seven fundamental principles of privacy by design in the context of their application to the healthcare sector [35].

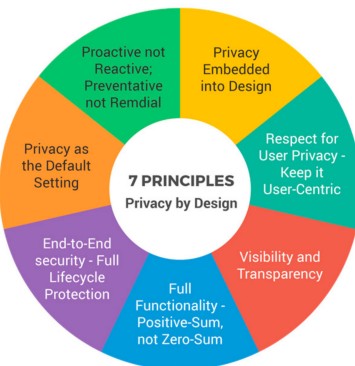

**Figure 2.** Seven fundamental principles of privacy by design [35].

### 2.4.1. Proactive Not Reactive; Preventative Not Remedial

This principle dictates that privacy and data protection are approached in proactive rather than reactive terms. In this method, privacy risks are anticipated and prevented before they occurred [35]. Such proactive privacy design is the standard for individual user participation in the electronic health records system, which heavily underlines the impact of proactively stopping negative health consequences [48].

### 2.4.2. Privacy as the Default

Privacy and data protection are automatically secured in every system as its default setup [47]. Thus, PbD guarantees the fact that personal data are protected automatically in all regular IT systems or business practice by facilitating the highest level of privacy and data protection [48]. In this way, the healthcare data is automatically protected as designed by default into the system [35]. This principle is mainly informed by the following FIPs:

- Purpose specification—the purposes of personal information collection, uses, retains, and disclosure needs to communicate the data subject before the collection of information. Hence, the purpose of information collection in a healthcare system needs to be lucid, within limit and relevant to the circumstances [35].
- Collection limitation—personal information collection in a healthcare system requires being fair, lawful, and limited to that which is mandatory to the specified purposes [35].
- Data minimization—personally identifiable information collection needs to be kept at a minimum wherever possible. Non-identifiable interactions should be set as the default in the healthcare information and communication technologies [35].
- Use, retention, and disclosure limitation—in a healthcare organisation, personal information should be retained only as long as necessary in order to fulfil the clear purposes and needs to be destroyed securely. Consequently, the use, retention and disclosure of healthcare information should be limited to the individual unless the person has consent, except if it is required by law [35].

### 2.4.3. Privacy Embedded into Design

This principle ensures integration of privacy through the development and implementation of a systematic program [35]. As a result, privacy is embedded in the architecture of a healthcare IT system without weakening its functionality. Such a program should be standards-based and responsive to review and validation [47]. In a healthcare system, privacy must be embedded into operations and information architectures in a comprehensive and creative way. A systemic method needs to be adopted that is amenable to external reviews. FIPs should be applied at each step in the design and operation of healthcare system. Moreover, privacy impact and risk assessments should be carried out and disclosed wherever possible; the privacy risks and measures selected to mitigate these risks need

to be presented clearly. The privacy impact of the healthcare technology needs to be minimised and should not simply break down through use, error, or misconfiguration [47].

### 2.4.4. Full Functionality—Positive-Sum Not Zero-Sum

PbD denies unnecessary trade-offs such as privacy vs. security or privacy vs. availability in order to gather all the interests and objectives [35]. This positive-sum model is increasingly important in the health technology industry. The key is to maintain a patient outcome focus without enforcing a trade-off between good health and privacy. The purposes for which the health data are accessed and by whom should be authorized by individuals' choices [48].

### 2.4.5. End to End Security—Lifecycle Protection

PbD has been embedded into the system before the first information being collected and extends to the end of the cycle, this means PbD applies throughout the whole lifecycle of the processing and ensures that data will be erased in a timely manner at the end of processing [47]. Accordingly, the feature of appropriately executed log data files in the healthcare sector is ensured by this principle by increasing the flexibility in order to establish the data quality principles and accountability [48].

### 2.4.6. Visibility and Transparency

Accountability requires that all technology and business practice involved in maintaining all actions is transparent and visible in order to assure every stakeholder [35]. Moreover, the ability to verify both protection and maintain the access audit logs to the health data is a key in instilling user confidence in the healthcare system [48].

### 2.4.7. Respect for User Privacy

Above all, a prominent criterion of the whole process to create PbD is to offer individuals, by default, a formidable privacy, proper notice, and options that are user-friendly [35]. This applies to health sector technologies that gather, apply, store, or manipulate personal data [47]. The EU General Data Protection by Design (GDPR) includes data protection by design and data protection by default, which are the foundational principles from Ann Cavoukian's privacy by design. Canada's privacy commissioner included these principles in 'Privacy, Trust and Innovation—building Canada's Digital Advantage' [49]. In Australia, the Victorian public sector adopted these principles as a cardinal policy to information privacy management practices. In 2012, the U.S. Federal Trade Commission identified privacy by design principles as their recommended practices for shielding privacy. Moreover, the U.K. information commissioner has highlighted the necessity of the privacy and data protection by design [49].

Principles of privacy by design are the core assumptions, whereas privacy design strategies are the commands that will apply to the overall planning and conduct of privacy in the information system. Dr. Jaap-Henk Hoepman derived the following eight privacy design strategies: minimise, hide, separate, aggregate, inform, control, enforce, and demonstrate [50]. IT architects will be able to incorporate privacy by design early in the software development lifecycle, primarily during the analysis and concept development phase with the aid of such privacy design strategies.

### 2.5. Privacy Design Strategies

Privacy design strategies are grouped into two parts—data-oriented strategies and process-oriented strategies. Approaches or strategies that are mainly data-oriented can easily relate to privacy by architecture approach as pointed out by Spiekermann and Cranor [51], whereas the process-oriented strategies define the privacy by policy approach.

2.5.1. Data Oriented Strategies

1.  Minimise: Minimise is the most elementary privacy design strategy, stating that only a minimal amount of personal data should be processed. This strategy is broadly discussed by Gurses et al. [52]. Accordingly, it is important to ensure that unnecessary data is not collected; thus, the probable impact relating to privacy of a system is marginal. One must express where the use of personal data is in terms with the purpose [19]. One can agree not to collect any data about a subject at all. On the other hand, he or she can agree to gather only a small group of attributes [53]. Design patterns: Design patterns that put this strategy in practice are "select before you collect" [54], and use pseudonyms and anonymization [55].

2.  Hide: This strategy basically points out that information that is personal in nature and its interrelationships must not be decipherable in plain sight. The logic behind this strategy exploits the fact that concealing personal data from plain view prevents a range of abuses [56]. It does not specifically say to whom the data should be hidden from. This relies on the exact situation in which this strategy will be actioned. Spontaneous appearance of information from the application of the system will be hidden by this strategy. In other cases, if the data are gathered, collected, or processed lawfully by one entity, the objective is to stash the data away from other parties. In this circumstance, confidentiality is ensured through this strategy. Design patterns: Design patterns within the boundary of "hide strategy" are several-fold. One such pattern is data encryption (in transit or stored, anonymization or pseudonyms), techniques that de-link some connected events such as attribute-based credentials. Data encryption is a security method where information is encoded and can be accessed only with the correct encryption key. It translates data into another form, and therefore a decryption key is required to access the information [55].

3.  Separate: This strategy states that personal data should be in separate partitions and, if possible, should be managed in a distributed fashion. By segregating the storage or processing of personal information of multitude of sources belonging to the same person, complete profile of the same person cannot be determined [57]. This strategy demands for distributed processing rather than centralised solution. Particularly data from distinct sources should preferably be stored in unconnected and different. Design patterns: No exact design pattern has yet been identified for this strategy [53].

4.  Aggregate: This strategy states that personal data should be managed with the least possible details and the maximum level of aggregation in which it is valuable. Aggregation of data restricts the amount of details in personal data over groups of attributes or individuals. Thus, this information becomes less sensitive. When the information is adequately uneven, the size of the group over which it is aggregated is quite substantial, and a scant amount of data can be attributed to a single person, resulting in the protection of privacy [53].

    Design patterns: Design pattern examples belong to this strategy are as follows:

-   Dynamic location granularity (normally deployed in location-based services) is presented by a device with real-time location. This design pattern supports the minimised data collection and distribution. It is important if a service is collecting location data and transferring this data to a third party. In order to provide some contextual services, many location-based services collect current location data from users. Unnecessary data collection may risk the service by harming the user's privacy. Accepting location data at different levels requires a location hierarchy by both services as well as a complex data storage model of more than simple digital coordinates [50].

-   K-anonymity: K-anonymity is a vital model that safeguards joining attacks in privacy protection. It is a property of a dataset that is used in order to describe the dataset's level of anonymity [58].

2.5.2. Process Oriented Strategies

5.  Inform: This strategy corresponds to the vital notion of transparency. If personal data is processed, data subjects should be adequately up-to-date. When user uses a system, they need to be

sufficiently informed about which data gets processed and the reason for such processing. This consists of information about the protection mechanism of data in question and the transparency on the security of the system. The user should be informed as to what information is shared to the third parties and they should also be notified about their rights regarding data access and ways to exercise those rights [53]. Design patterns: Both platforms for preferences related to privacy and data infringement notifications are design patterns falling into this class. Graf et al. provided an intriguing assortment of privacy design patterns to inform the user from the perspective of human computer interfacing [59].

6.  Control: The control strategy defines that the user should be equipped with the necessary measure over the processing of their personal data. This strategy is a critical corresponding part to the "inform strategy". There is little use to inform the user that the personal data is collected without practical capability of controlling the usage of one's personal data [60]. Users often get the right to update, view, and even ask to delete personal data that is collected by the data protection legislation. This strategy emphasizes this fact, and this pattern gives users the tool to exercise their data protection rights [53]. Control empowers the user to decide whether to use the system and control the kind of information about them getting processed [61]. However, this strategy surpasses the strict application of rights corresponding to data protection. Likewise, providing users the control over their personal data will mostly result in error correction. As a result, the processed personal data quality may increase. Design patterns: No exact design patterns identified that can fit the strategy [53].

7.  Enforce: Enforce defines that privacy policies that are harmonious with legal obligations have to be in place and must be enforced. Enforce ensures that a privacy policy is rightly and properly in place. This strategy ensures that the system respects privacy during its operation. More importantly, the policy needs to be enforced. To prevent the violation of the privacy policy, appropriate technical protection measurements are established. Moreover, policy must be established by an appropriate governance structure [50]. Design patterns: This strategy is implemented by the design patterns such as access control. Another example is policies and privacy rights management—license to personal data including the digital rights management form [53].

8.  Demonstrate: The final strategy defines the connection of a data controller to control compliance with privacy policy and applicable requirements. The strategy entails the data controller to demonstrate that it is in control, and therefore this is a step ahead of the "enforce" strategy. In case of complications, the extent of any probable privacy infringement should immediately be measurable by the user.Design patterns: Design patterns implementing such strategy are, for example, the use of logging and auditing, as well as the privacy management system [59].

These privacy design mechanisms are not only beneficial while developing systems that are privacy friendly, but also aid in gauging the privacy impact assessment of the IT system. To implement privacy by design in a system the most impactful tool that can be deployed are "privacy impact assessment" and "security incident management", which are described below.

*2.6. Privacy Impact Assessment (PIA)*

Privacy impact assessment (PIA) is a systematic assessment of a project that identifies risks of individuals' privacy. PIA analyses the risks and recommends solutions for managing, minimising, or eliminating their impact in the form of privacy control [8].

PIAs are important factors in privacy protection and should be part of the overall risk management and planning process. Undertaking a PIA can assist in the following:

*   Defining how personal information flows in a project.
*   Investigating the possible impact on individuals' privacy.
*   Classifying and recommend options for avoiding or minimising negative privacy impacts.

- Constructing the privacy considerations into the design of a project.
- Achieving the goal of the project while enhancing the positive and minimising the negative privacy impact [40].

Success of projects depends on whether they meet legislative privacy requirements and the privacy expectation for the community. If the privacy issues are not properly determined, it can impact the community's trust in an entity and challenge the project's success. Risks of not undertaking a PIA include:

- Non-compliance with the spirit of relevant privacy laws, potentially leading to a privacy breach.
- Loss of credibility through lack of transparency in response to public concern about handling personal data.
- If the project fails to meet expectations, it can damage an entity's reputation about how personal data will be protected.
- In a project development or implementation, if the privacy risks are identified at a late stage, it can result in unnecessary costs or inadequate solution [9].

Seven Core Elements

To achieve an effective PIA, seven core elements could be utilised [8,9,62] as follows:

1. Integral to an organization's governance: Governance structure of the organization should be integrated with the PIA, with a clear direction on who has obligation over the PIA.
2. Fit for purpose: PIA should be proportionate with the probable privacy risks associated with the project.
3. Comprehensive: PIA should not only cover information privacy but also all privacy issues. PIA should consider if any modification is required in secondary documentation such as human resource policies and privacy management plans.
4. Available: The report of the PIA should be publicly accessible. If not possible, releasing a summary report of the PIA to notify and search for feedback on privacy issues should be considered.
5. Enables compliance: PIA must address all privacy obligations, including health privacy principles (HPPs) and information protection principles (IPPs), where applicable.
6. Ongoing: PIA should cover a constant review mechanism in order to evaluate privacy issues throughout the life cycle of the project.
7. Constructive: PIA should support the privacy values of the organization and reference the organization's risk management process [9,62].

Organisation must not disclose personal information for the purpose of marketing unless the individual has consented. Wider opportunities for action may be identified by the PIA. It may identify that there are parts of the business where PIA might help to achieve better security and better accuracy. PIA helps to take reasonable steps in order to protect personal information from loss, misuse, and unauthorised access. PIA is supportive by ensuring privacy compliances well as identifying better practices [8,62].

*2.7. Security Incident Management (SIM)*

Strong security is imperative to a reliable privacy preserving framework. Without security incident management, personal data can simply fall into the wrong hands. Personal information security infringements are rather troublesome to remediate effectively [37].

SIM supports organizations in achieving events pertaining to information security, related incidents, or vulnerabilities. SIM offers an instant response to security occurrences in a method that shields individuals who have been affected, meets officials' expectations, and eventually results in preserving the organizational reputation. For organizational information security management, SIM is a component of broader requirement [37].

Four Key Stages of SIM

Security incident management (SIM) consists of four key stages (Figure 3) [37,49]:

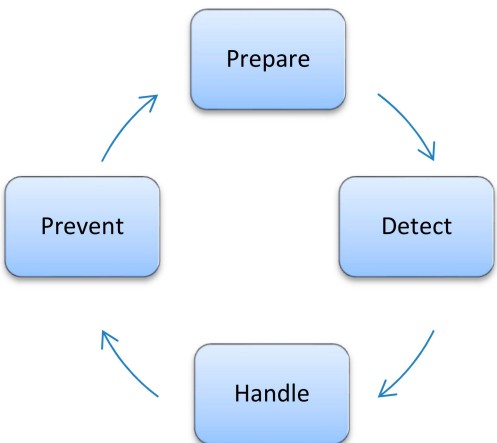

**Figure 3.** Four key stages of security incident management (SIM) [37].

1.　Prepare: To handle incidents, for example, formulate an incident management regulation policy and create an expert team to manage the incidents.
2.　Detect: Detect and report data security breaches.
3.　Handle: Measure incidents, and decide ways to tackle and respond to those incidents.
4.　Prevent: Internalize the lessons, meaning instead of just finding out how things could have been completed in a better way, it demands making necessary alterations to enhance the process [49].

## 3. Critical Analysis on Privacy by Design Frameworks

Various privacy by design frameworks are available to shield personal information. Between these, five frameworks have been selected because they are proposed for healthcare information systems and other environments for protecting personal information. Moreover, three of the frameworks are implemented and tested in information system environment.

### 3.1. Privacy by Design in Victorian Public Sector

New technologies are embedded into everyday lives of people and functionalities of private and public sectors; thus, technologies mould the expectation of a community in a government. This framework provides context about privacy by design, a record of its primary features, and defines the reason for it being beneficial to the community and for the Victorian public sector organization. This framework allows privacy to be embedded or somewhat 'built into' the design and architecture of information systems, networked infrastructure, and business process. The aim of this framework is to make sure privacy is projected or thought of beforehand at the beginning and within the complete duration of the development and implementation of initiatives that involves the collection and management of personal data [47,63].

One of the features of this framework is adding public value with individual right. Though privacy is deemed as an individual right, an extra community dimension has been included to it by identifying the fact that privacy has an important decisive impact on the formation of public value. The construction of public value by the public sector relies on information systems and business processes that process and collects data, usually personal data. To integrate them into the public sector, modernization is a challenge with regards to respecting the privacy and production of public value. OECD described the public value by the following six board examples [34]:

- Belongings and services meeting the expectations of residents and clients.

- Development choices that surely satisfy citizens' expectations of fairness, justice, effectiveness, and efficiency.
- Fairness and productivity of distribution.
- Appropriately synchronized and results-oriented public institutions that reflect citizens' preferences and desires.
- Legal application of resources to attain public purposes.
- Innovation to change preferences and requests.

This framework aligns with the methodology through enabling privacy to be embedded in the business process and information system, not only as productivity measures but also to come across public expectations about the use of personal information in the government [49]. This framework supports project management and good governance, and also promotes costs savings in ICT-enabled projects and offers the least invasion into the privacy of individuals. Organizations can implement this framework from either a full rollout approach or a project-by-project style. This framework offers effectual privacy management, particularly in the Victorian public sector. However, there are still some limitations, such as inefficiency in maintaining the reliability of personal information. Secondly, there is no clear direction about the management and control of personal information. Therefore, the confidentiality of personal information gathered and used by the Victorian public sector may be compromised rather easily.

### 3.2. Framework for the Design of Privacy Preserving Pervasive Healthcare

Privacy is a significant part of global computing systems, particularly the global healthcare sector [64]. This framework was constructed as a common framework for the design of privacy protective general healthcare. The aim of this framework is to remove large obstacles in order to deploy pervasive healthcare systems and acceptance of technology by addressing the issues. It specifies the design of a system that intends to minimize the privacy impact on developing privacy sensitive healthcare applications. This framework combines a number of methods proposed by several researchers that address privacy in healthcare applications. The methodologies are outlined as follows. Moncrieff et al. [65] proposed the use of indicators to determine the environmental context of activity within the application environment. Wickramasuriya et al. [11] used contextual identity to determine whether an individual is authorized to use the system. Tentori et al. [64] proposed "user preferences" according to contextual information and "data filter" that determines the right privacy policy to apply as per given preferences and the context of the environment or the applicable rules. Alternatively, the "context" and the "preference" adjust the privacy with respect to the user. In an effort to combine the approaches and to formulate a more formal framework for designing privacy in pervasive healthcare application, this framework extract and combined foundations such as policies and principles are used to combine distinct approaches into one entity and produce a more formal methodology by reviewing existing approaches.

This framework represents and proposes a design framework used to achieve its goal and will be built in with the information process flow [66]. The user of the environment handles the "user preference", whereas the "rule" set indicates the effects of the user on privacy data filter. By allowing the privacy filter to be modified according to situation taking place within the environmental boundary, the context is incorporated into the privacy system. Both control and feedback are incorporated to reduce the leak of data. However, balancing of risk with trusts and benefits are forms of design factors that need to be considered when implementing ubiquitous computing [66]. These factors are not commonly exclusive to the design framework, but they nonetheless are aspects that need be considered in implementing instances of the framework [67,68]. In addition, the application and the outcome of the protection of data privacy is not acknowledged. This framework supports the implementation of privacy measures in a ubiquitous computing environment and has demonstrated its application to pervasive healthcare. Additionally, it is constructed on the basis of existing methodology; particularly

data filter technique, though it is not clearly defined if this framework is based on any proven method such as whether any standards, tools, or principles by privacy by design have been incorporated. The sensitivity of the healthcare environment and its related data are another restriction that can have a huge impact in the adoption of this framework.

### 3.3. PRIPARE

Making the industry to adopt privacy by design (PbD) by backing its applications in a research is referred to as "PReparing Industry to Privacy-by-design by supporting its Application in Research (PRIPARE)". This framework was established in the European Union for technological development, research, and demonstration with two-fold mission. Firstly, it was established to facilitate the information system with privacy and security by design methodology, as well as supporting its practice to the ICT community. Secondly, it was developed to foster risk management culture through educational material targeted at stakeholders [21,69]. PRIPARE proposed a methodology for the application of PbD that can be easily combined with most system development phases. PRIPARE captures and integrates from the existing standards, practices, and research proposals on privacy engineering. In this framework, privacy principles were directly extracted from the EU General Data Protection Regulation (GDPR). Principles are then specified into guidelines. However, International Organization for Standardization (ISO) 29100 privacy principles are used to illustrate the operationalization process. The proposed PbD process is divided into seven phases: analysis, design, implementation, verification, release, maintenance, and decommission [70]. Environment and infrastructure is an additional phase that is a central item that deals with organizational structure. The ISO 15288 standard is used for mapping how the seven phases may easily be adapted to the lifecycle of an organization (Table 2) [71].

**Table 2.** International Organization for Standardization (ISO) 15288 for mapping PReparing Industry to Privacy-by-design by supporting its Application in Research (PRIPARE) phases [71].

| PRIPARE Phases | ISO 15288 System Lifecycle Process |
| --- | --- |
| Environment and Infrastructure | • Project privacy portfolio management process<br>• Infrastructure management process |
| Analysis | • Privacy requirement analysis process |
| Design | • Privacy architectural design process |
| Implementation | • Privacy implementation process |
| Verification | • Privacy verification process |
| Release | • Transition process |
| Maintenance | • Maintenance process |
| Decommissioning | • Disposal process |

PIA process is integrated to run in parallel at the analysis phase. The analysis phase consists of the following processes such as legal assessment, detailed privacy analysis, functional description and high-level privacy analysis, privacy and security plan preparation, operationalization of privacy principles, and risk management. The operationalisation of privacy principles aims to replace technical observable measures with abstract privacy principles. In this process, privacy principles and guidelines need to be selected and refined into a set of detailed privacy conformance principles that will outline organisational and technical requirement. PRIPARE's available mechanism needs to be more productive and detailed for an IT system that will be an optimistic move towards operationalizing privacy by design.

### 3.4. Enhanced E-Health Framework for Privacy in the Healthcare System

Electronic healthcare recording has now become an important aspect of maintaining patients' medical data. The data are stored within a certain infrastructure and the framework around the accommodating electronic health record (EHR) is implemented to build a safe and secure system, with privacy of patients' health information in the health industry as the main priority [72,73].

To secure the healthcare data from unauthorized users, a superior framework called the multi authority-based encryption (MA-ABE) encrypted technique has been put forward. [74]. MA-ABE enhances system scalability as well as aids in attaining fine-grained access control. Moreover, outsider attack, such as man-in-the-middle attack or eavesdropping denial of services, is managed efficiently in this encryption method. There are both positive and negative impacts through the analysis of SPSS [75]. In this framework, there is a feeble relationship between privacy and negative impact, for example, patient information can be accessed by the administrator for stealing data [76]. To overcome this, patient information can be accessed only by doctor's consent or by patient's consent. Moreover, cloud computing involvement is also weak, as anyone can access data because it is a third-party service [77]. Therefore, while hoarding data in the cloud, advance encrypted standard (AES) needs to be used. To improve the privacy and security of the personal health record, AES is the latest encrypted technique. Hence, this framework needs to use the multi authority-based encryption (MA-ABE) methods for securing PHR data with advance encryption standard (AES), as well as needing to discover how SPOC (single point of contact) supports in gaining benefit in health security [78]. This framework addresses the issue in privacy such as unauthorized users accessing sensitive data from a patient's health record, which should be hidden. Patients' health records are often outsourced for storage at a third party. In this framework, access control scheme and patient-centric personal data with enhanced encrypted satisfaction method needs to be considered. Additionally, hash-based digital signatures [79] and pseudo-identity need to be used to identify the privacy of personal data [80]. Moreover, it addresses the enhanced privacy model of additional authorization and authentication of functionality and discovers the novel strategies that need be deployed to gradually develop the efficiency on privacy and user in the e-healthcare system. The data need to be analysed using SPSS tool or a survey to test the e-health framework [81,82].

### 3.5. Privacy by Design Framework for Assessing Internet of Things Applications and Platforms

Internet of Things (IoT) are designed as well as developed as impartial applications, either from scratch or with the necessary expertise of IoT middleware platforms. This framework was created on the basis of Hoepman's privacy design strategies and Ann Cavoukian's seven fundamental PbD principles. This framework is evaluated by a few open source IoT middleware platforms, namely, Open IoT and Eclipse Smart-Home. It has a set of guidelines that can be used to evaluate privacy competences and gaps of current IoT applications, as well as middleware platforms [83,84]. The development of this guideline supports software engineers in implementing this guideline in a customized manner into their IoT application. This framework does not discourage such approaches if data are developed over proper consent processes. However, IoT applications need to take all possible actions to achieve their goals with the smallest amount of data, particularly medical data. It is presented as a conceptual framework that integrates PbD principles in the systematic assessment of the privacy capabilities of healthcare IoT applications and platforms in order to guide software engineers. This framework supports the assessment of open source IoT platforms and efficiently provides step-by–step processing of the use of this framework [85]. Moreover, no impact assessment tool is integrated to measure the impact of this methodology (Table 3). Thus far, IoT applications and middleware platforms did not consider the privacy concerns explicitly. This is comparatively due to the paucity of systematic methods aimed specifically at designing privacy that can direct the software development procedures in IoT.

**Table 3.** A comparative view on existing privacy by design frameworks.

| Privacy by Design Key Parameters | Frameworks | | | | | | |
|---|---|---|---|---|---|---|---|
| | 1 | 2 | 3 | 4 | 5 | 6 | 7 |
| **Seven Fundamental Principles of PbD (Ann Cavoukian)** | PbD in Victorian Public Sector [34,47,49,63] | Framework for the Designing of Privacy Preserving in Healthcare [11,64–68] | PRIPARE [65,71] | Enhanced E-Health Framework for Privacy in Healthcare System [8,72,75,76,78,80,81] | PbD Framework for Assessing IoT Applications [84,85] | PISCES [86,87] | ISO/IEC 29110 with PbD in Health Care Sector [73,88,89] |
| 1. Proactive not reactive; preventative not remedial | √ | | | √ | √ | √ | √ |
| 2. Privacy as the default | √ | | | √ | √ | √ | √ |
| 3. Privacy embedded into design | √ | | | √ | √ | √ | √ |
| 4. Full functionality—non-zero positive-sum | √ | | | √ | √ | √ | √ |
| 5. End-to-end security—full lifecycle protection | √ | | | √ | √ | √ | √ |
| 6. Visibility and transparency—keep it open | √ | | | √ | √ | √ | √ |
| 7. Respect for user privacy—keep it user-centric | √ | | | √ | √ | √ | √ |
| **PbD Strategies (Hoepman Jaap-Henk)** | | | | | | | |
| 8. Data-oriented strategies: | | | | | | | |
| 9. Minimise | | √ | √ | | √ | | √ |
| 10. Hide | | √ | √ | | √ | | √ |
| 11. Separate | | √ | √ | | √ | | √ |
| 12. Abstract | | √ | √ | | √ | | √ |
| 13. Process-oriented strategies: | | | | | | | √ |
| 14. Inform | | √ | √ | | √ | | √ |
| 15. Control | | √ | √ | | √ | | √ |

**Table 3.** *Cont.*

| Privacy by Design Key Parameters | Frameworks | | | | | | |
|---|---|---|---|---|---|---|---|
| | 1 | 2 | 3 | 4 | 5 | 6 | 7 |
| Seven Fundamental Principles of PbD (Ann Cavoukian) | PbD in Victorian Public Sector [34,47,49,63] | Framework for the Designing of Privacy Preserving in Healthcare [11,64–68] | PRIPARE [65,71] | Enhanced E-Health Framework for Privacy in Healthcare System [8,72,75,76,78,80,81] | PbD Framework for Assessing IoT Applications [84,85] | PISCES [86,87] | ISO/IEC 29110 with PbD in Health Care Sector [73,88,89] |
| 16.   Enforce | | √ | √ | | √ | | √ |
| 17.   Demonstrate | | √ | √ | | √ | | √ |
| **Privacy Impact Assessment (PIA)** | | | | | | | |
| ➢    Integral | √ | | √ | | | √ | |
| ➢    Fit for purpose | √ | | √ | | | √ | |
| ➢    Comprehensive | √ | | √ | | | √ | |
| ➢    Available | √ | | √ | | | √ | |
| ➢    Enables compliance | √ | | √ | | | √ | |
| ➢    Ongoing | √ | | √ | | | √ | |
| ➢    Constructive | √ | | √ | | | √ | |
| **Security Incident Management (SIM)** | | | | | | | |
| ➢    Prepare | √ | | | √ | | | |
| ➢    Detect | √ | | | √ | | | |
| ➢    Handle | √ | | | √ | | | |
| ➢    Prevent | √ | | | √ | | | |
| **Public Value by OECD** | | | | | | | |
| Additional community dimension included to privacy | √ | | | | | | |

### 3.5.1. IoT Privacy Requirements

Privacy for IoT devices (especially related to healthcare) can be further fortified by firstly understanding the necessity of IoT technologies and improving the technologies related to privacy issues. Secondly, it can be done by researching on state laws, if found, to regulate the operations of IoT and thus create a framework for these laws where and when to be applicable to protect privacy and the data flow [90].

According to a survey conducted by Trend Micro Inc. in 2016, 44% of the surveyed individuals were worried about their privacy [91]. Only the demand of consumers for secure IoT devices can alter the attitude of major IoT device vendors to formulate a secure configuration [92]. Various challenges faced by Internet of Things in cases of privacy such as protecting the location of the person from the associated device, protecting very sensitive personal information through monitoring the IoT devices used, localising the data as much as possible by using a decentralised authentication key management, and by restricting the amount of non-essential data that are needed for user identification [93].

Under the security and privacy needs, the following requirements have been pointed out [94]:

- *Resilience to Attacks:* The device should be able to withstand an attack over the network. It should fine-tune itself after any device failure and accordingly re-start services.
- *Data Security:* All data entering and leaving the network must have proper authentication.
- *Data Access Control*: Information providers should have tight control over the flow of data.
- *User Privacy:* Only the provider should have access to user data, and this should be kept under strict guidelines for providing services. Figure 4 shows the major security concerns for IoT devices.

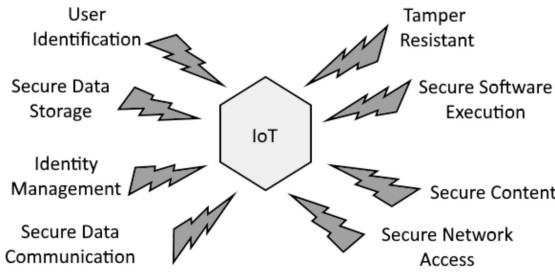

**Figure 4.** Major security concern for Internet of Things (IoT) [95].

### 3.5.2. Security Threats and Privacy Requirements for SOA-based IoT Middleware and Industrial Impact

Previous studies have shed light on how the middleware system has seen an exponential growth from simply hiding network protocols to handling more complex tasks such as communication between two network devices to handling data and managing security. In addition, multiple challenges have been confronted by the service-oriented architecture (SOA), and security is one of the critical challenges that strongly demands for a security architecture standard based on service-oriented architecture to safeguard the data. There is a budding thought that innovation, competitiveness, and creativity must be addressed from a "design-thinking" perspective, that is, an avenue to view the world and resolve constraints that is holistic, integrative, interdisciplinary, inspiring, and innovative all at once.

Privacy should also be approached from a similar angle of a design-thinking perspective. Privacy needs to be ingrained into the networked data systems and infrastructure, by default. Privacy must become inherent to various critical organizational priorities, design processes, project objectives, and operational planning. Privacy must be incorporated into every standard, process, and protocol that touches our everyday lives. This research attempt intents to materialize this possibility by establishing a universally general framework for the strongest protection of privacy that is currently available [35].

The manufacturing industry at present is one of the most commonly hacked industries, eclipsed only by healthcare, as highlighted by IBM's 2016 Cyber Security Intelligence Index. The vulnerabilities are often found in businesses believing that they are not probable targets because they do not have any

significant or large amount of consumer data and, therefore, they are not interested in investing any resources on cybersecurity. However, recent happenings show that the industry is far from immune; such as in 2019, where it was detected by the researchers at Kaspersky Lab that a sophisticated cybercrime operation was already in motion to destructively affect at least 130 manufacturing, industrial, and engineering firms across the globe. Named "Operation Ghoul", it used email phishing techniques [79] to spoof letters from banks to make unsuspecting recipients reply them with highly sensitive corporate information [96]. Email phishing is one of the most common ways of carrying out spam attacks on senders, and is achieved through manipulating different email header and body fields [79].

With the sharp increase of the application of IoT-based devices in manufacturing, the impact on security is complicated even further. For instance, how dangerous will it be in cases where a BMW customer's car is sending over data on real-time diagnostics, allowing remote control and potential hacking of the car itself? It is now no longer traditional computers that are the gateway to a business establishment; rather everyday things such as cars, thermostats, and other home appliances all need to be considered. Thus, privacy by design (PbD) considerations in manufacturing industries of all types are imperative in the times that we are living in [96].

### 3.6. PISCES

The ground of this framework was established from Cavoukian's seven fundamental principles of PbD. Privacy incorporated and security enhanced system (PISCES) is a framework that aims to establish foundational pillars for implementing privacy or security by design pertaining to Internet of Things (IoT). This framework is a pioneering approach that actions to (1) verify the source of the privacy-sensitive data (2) control the path of data throughout their application, and (3) provide the right to own these data to other third parties. In the parlance of privacy protection and data protection, PbD needs to be considered at the initial stage of product design. This framework functions as a stringent separation between data controller and provider, where providers manage the data privacy and controllers are accountable for privacy and protection of the provided data [86]. The separation of the roles is guaranteed by the Controller Smart Data System (CSDS) that is established from the Smart Data System (SDS). It processes data along with its privacy settings (metadata), offering the options of private management of data defined by the users [97]. Against the need to access information, SDS also balances user privacy in the case of activity, for example, law enforcement agency such as police investigation against crime. Building a privacy validation chain (PVC) gives permission to the data owner and/or users (data controllers, data users) to identify to whom and for which purpose the data will be used [98]. This framework is believed to obtain a practical reduction for internet service providers and users when monetizing user data. It makes the explanation of fair and mutually acceptable conditions for using the services and data compulsory [99].

To protect privacy by eliminating or reducing personal information or by preventing undesired and unnecessary data barring the loss of the functionality of the information system, PET act as a coherent system [100,101]. This framework is governed by three main policies (1) data separation through Smart Data System, (2) combat against crime, and (3) rational data monetization. However, the shortcomings underlined the fact that the legal framework fell short to ensure the protection of the private sphere [102]. In regards to the privacy protection of ICT systems, this framework requires the establishment of the proven fundamental principles in a right tactic to directly integrate privacy preserving from the initiation of the system and throughout the operation of the system. Likewise, it encourages the use of privacy management when necessary.

### 3.7. Extending ISO/IEC 29110 Basic Profile with Privacy by Design Approach: A Case Study in the Health Care Sector

The most commonly encountered goals of privacy and privacy addressing practices were taken into account to construct the framework for the healthcare sector. This framework describes an incorporation of PbD goals into International Organization for Standardization/International

Electrotechnical Commission (ISO/IEC) 29110, targeted at small software development organisation. Ann Cavokian's seven fundamental principles of privacy by design are included as a guide for the deployment of this framework. In addition, the top 10 of the most frequently mentioned privacy goals by Morales-Trujilo et al. are used as a basis to generate privacy design strategies and tactics. ISO/IEC 29110 is primarily aimed at Very Small Entities (VSEs) that are based on ISO/IEC 15289, ISO/IEC 15504, and ISO/IEC/IEEE 12207 [103,104]. In this framework, the ISO/IEC 29110 basic profile aims at enhancing the quality of software systems by guiding project management (PM) and software implementation (SI). The PM process helps to identify and guide the activities of software development activities. Alternatively, SI process controls the analysis, design, construction, integration, and testing activities during the software development project. This framework considers the importance of defining the software development process in order to manage sensitive data adequately. Consequently, this framework proposes a solution of integrating PbD-related tasks, as well as the role of ISO/IEC 29110 basic profile [105]. The fundamental principles of privacy by design and privacy strategies are integrated with the international standard ISO/IEC 29110 to design the framework; however, there is no direction of privacy impact assessment or security incident management (Table 3). The results of using this framework in a healthcare system cannot be generalised, as more thorough validation testing needs to be employed [88,89]. A comparative summarized view is presented in Table 3.

## 4. Conclusions and Future Work

After a detailed analysis, this study produced several observations, primarily in the field of privacy by design. The literature review presented in this paper discussed information including substantive findings and theoretical and methodological contributions that are evaluated to identify problems. Existing privacy by design frameworks were critically analysed and a comparative analysis is presented (Table 3) to identify the limitations that are being used by scammers to conduct data breaches. It was also underlined in this study that certain fundamental components are missing in existing privacy by design frameworks, such as the seven fundamental principles by Ann Cavoukians, PbD strategies, privacy impact assessment (PIA), security incident management (SIM), and public value by OECD (Organisation for Economic Co-operation and Development). We also came to the conclusion that these components are quite generic, and thus the potentiality of developing the research into a hybrid framework is quite promising. Moreover, the contemporary practice of dealing with a data breach is not the most effective approach, as has been pointed out, and therefore requires a more comprehensive methodology that will consider the various perspectives of the problem. This review paper discussed the key contexts of privacy by design and highlighted the key characteristics of existing frameworks. The shortcomings of each of the frameworks were also clearly mentioned in Table 3. It is clear from this review that current solutions are still unsuccessful in delivering expected value, as the rate of data breach continuing to increase. The privacy by design framework should embed the key parameters in any future development, which is highlighted in Table 3. In future, a refined and improved framework can be designed by including all these parameters into system architecture in order to reduce the rate of data breaches mainly occurring in the healthcare sectors. Open source data, for instance, OpenMRS [106], Freehealth.io [107], and ZHHealthcare [108] have been collected, and in future they will be used to develop and test the proposed framework. The data collection and data storage method for testing will be the dynamic sector of this study. Unfortunately, in spite of several considerations from multiple private organisations and government bodies of leading nations of the globe, we have fallen short in forming effective methodology that can really have a deciding impact on the data breach issue. However, the steps to reinforce data privacy have seen substantial significance in recent times, resulting in more research and ample availability of funding in this field. Hence, we can expect that a fortified and reliable framework against the highlighted limitations in this study will benefit in reducing the data breach globally.

**Author Contributions:** Conceptualization, F.H.S., S.A., B.S. and K.C.Y.; methodology, F.H.S. and S.A.; validation, S.A. and B.S.; writing—original draft preparation, F.H.S.; writing—review and editing, S.A. and B.S.; visualization,

F.H.S. and S.A; proof reading, B.S and K.C.Y. All authors have read and agreed to the published version of the manuscript.

**Funding:** This research received no external funding.

**Conflicts of Interest:** "We are the authors of "A Systematic Literature Review on Privacy by Design in the Healthcare Sector" we declare that no conflict of interest."

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
