# Peer review of "A Systematic Literature Review on Privacy by Design in the Healthcare Sector"

_electronics, doi:10.3390/electronics9030452_

Round 1

Reviewer 1 Report

This english writing of this paper is pretty good, and I believe authors have investigated a lot of references regarding the data privacy in healthcare field. However, I do not think this paper is ready to be published in the current shape. Let me illustrate some major issues below.

  1. To be honest, this paper is more like a tutorial of seven fundamental principles proposed by Cauvokians than a survey. A survey needs to present the technical roadmap and indicate the future research trend. I have not seen both of them in this paper.
  2. This paper is called a systematic literature review, and I suppose it is the novelty of this paper. However, no related work has been included in this paper. Is this paper the first one to review the relevant publications in healthcare sector? If not, what is the difference between this work and the previous ones? This information is very important to determine the novelty.
  3. Usually, a literature review will compare the pros and cons of different methodologies, but I cannot find any in this paper. 
  4. What is the unique characteristics of data privacy issue in healthcare sector? It seems to me it shares most critical ones of the common characteristics of data security and privacy. The reason why I ask this question is that we need to determine whether it is necessary to do a special literature review especially for this field. A general survey may cover all important findings.  We need this information to determine the contribution of this work.

Author Response

Reviewer # 1, Concern # 1

Concern: To be honest, this paper is more like a tutorial of seven fundamental principles proposed by Cauvokians than a survey. A survey needs to present the technical roadmap and indicate the future research trend. I have not seen both of them in this paper.

Author Response: We highly appreciate the time you have taken to review our work and provide a number of critical insights that have indeed made this research initiative a more rigorous and complete one. The point you have made about the nature of a survey paper is absolutely correct and indeed an indispensable part.

To address the concern, The Future Work section (Section 4, at page 16, line 749) now discusses a detailed technical road-map based upon the informative tabulation done in Table 3 (page 15). In summary, what we have observed from our deep dissection of existing policies that any forthcoming development in this area needs to address all the shortcomings found in the current frameworks (as pointed out in Table 3), and software systems should notch up the level of user-experience through both interactive user involvement and customization, as well as a measured limitation on the flow of information.

Reviewer # 1, Concern # 2

Concern: This paper is called a systematic literature review, and I suppose it is the novelty of this paper. However, no related work has been included in this paper. Is this paper the first one to review the relevant publications in healthcare sector? If not, what is the difference between this work and the previous ones? This information is very important to determine the novelty.

Author Response: Thank You for pointing out this important aspect of a Systematic Literature Survey. Indeed there are number of papers available related to the topic in addition to the ones we have discussed.

In response to your critical suggestion, we have added a thorough technical analysis of a number of relevant work in this field; and can be found under section 1.4, at page 6, starting from line # 89.

Besides, critical reviews regarding privacy issues have been done before as we have pointed out, but our study is basically a first of its kind when it comes to narrowing the issue of privacy right down strictly to the domain of ‘HealthCare’ only.

Needless to say, such a section on relevant studies were rightly required and this suggestion of yours impacted the completeness aspect of the paper quite decisively.

Reviewer # 1, Concern # 3

Concern: Usually, a literature review will compare the pros and cons of different methodologies, but I cannot find any in this paper.

Author Response: Thank You again for underlining one of most important aspects of a review paper. The newly added section (section 1.4) contains a detailed studies and dissection of several relevant studies done by a number of researchers. We have tried our best to analyse both the pros and cons of different frameworks and methodologies discussed in those studies, and indeed uncovered novel insights and information on the topic of privacy preservation that highly enriched our own knowledge in the field and the overall weight of this review paper.

Reviewer # 1, Concern # 4

Concern: What is the unique characteristics of data privacy issue in healthcare sector? It seems to me it shares most critical ones of the common characteristics of data security and privacy. The reason why I ask this question is that we need to determine whether it is necessary to do a special literature review especially for this field. A general survey may cover all important findings.  We need this information to determine the contribution of this work.

Author Response: We also acknowledge the issue that you have raised and yes a lot of commonality between general data privacy and data privacy in healthcare can be found as you have correctly mentioned. But we felt due to the following unique reasons, a data privacy in health care sector requires a field of study on its own:

  • Health data are highly sensitive and extremely personal in nature which is not always the case for other domain. The degree of sensitivity of data widely fluctuates in accordance with its domain.
  • Data breaches in health care sector are rising rapidly in comparison to any other sectors and we felt it is necessary to separately investigate why such rapid rise is observed in recent times especially and what are the gaps in the framework that are enabling such continuous act of information theft.
  • Furthermore, the degree of ethical and legal implications associated with healthcare data privacy is one of the highest in contemporary times. Thus addressing privacy concerns in this field, as we think, is important than a general policy on data privacy frameworks and guidelines.

============

Each of your comments and suggestions, as we have found after implementation, made a massive footprint in terms of making the paper more focussed, complete, structurally sound and informative. We sincerely appreciate the time you have taken to provide such a constructive and detailed review of our work.

____________________________________________

Reviewer 2 Report

  1. In 1.3, how do you ensure that these 253 papers are the most significant studies in the field of PbD. How do you select the keywords? Can you provide more details in this section?
  2. You downloaded 106 primary papers and 147 secondary papers. However, there are only 103 papers in your references section. Where are other papers? Please discuss the reasons.
  3. For me, it looks like the flow of the paper is not organized well. The sections are not connected to each other. I could understand that your sections are written from the mind map, but there should be a more clear way to discuss these components. Please carefully read your work again.
  4. Although you pointed out several limitations, these are not addressed and highlighted clearly. What can the audiences learn from your literature review? Any practical implications. Please address my concerns in your conclusion and future work section.
  5. Figure 1 is very unclear. Please replace it with a high-resolution picture.

Author Response

Reviewer # 2, Concern # 1

Concern: In 1.3, how do you ensure that these 253 papers are the most significant studies in the field of PbD. How do you select the keywords? Can you provide more details in this section?

Author Response: Thank you for you review comment on keyword selection, as you have pointed out, indeed keywords are a key component in determining relevant research on a field.

The primary method we have used for resource selection based on keywords - is a policy of ‘Maximum aggregated likelihood’, whereas we have selected the keywords we deem closely matches to the field of the study undertaken in this review; and ‘Only’ select those journal papers and articles that used the most number of keywords from the set of keywords that we have defined to begin with.

Reviewer # 2, Concern # 2

Concern: You downloaded 106 primary papers and 147 secondary papers. However, there are only 103 papers in your references section. Where are other papers? Please discuss the reasons.

Author Response: Thank You for pointing out this critical inconsistency.

In addition to the existing literatures, we have also analysed few others, bringing a total count of resources to 108, and has been updated correctly in relevant sections.

Reviewer # 2, Concern # 3

Concern: For me, it looks like the flow of the paper is not organized well. The sections are not connected to each other. I could understand that your sections are written from the mind map, but there should be a more clear way to discuss these components. Please carefully read your work again.

Author Response: We highly appreciate such an insightful remark, and indeed, upon a closer look we did in fact observed that oftentimes the continuity and logical flow are missing which may cause varying degree of hindrance to the reader’s attempt to internalize the findings derived out of the study.

To implement your valued suggestion, we have aggregated subsections from different parts of the paper, added new analysis and grouped accordingly. An instance of this can be observed through the placement and impact of section 1.4 - ‘Relevant Studies’. While, in section 3.5 – discussing how privacy affects IoT and industrial manufacturing, we have broken down previous contents into new subsections, added more relevant information on it and structured the overall section in such a way so that it maintains a consistent and natural flow to and from the bordering sections. The other paragraphs and sections have also been necessarily modified to reflect the constructive feedback that you have given, bringing in quite a positive change in terms of the structure and readability of the paper.

Reviewer # 2, Concern # 4

Concern: Although you pointed out several limitations, these are not addressed and highlighted clearly. What can the audiences learn from your literature review? Any practical implications. Please address my concerns in your conclusion and future work section.

Author Response: Thank you for the excellent recommendation. The Future Work section (Section 4, at page 16, line 749) now discusses a detailed technical road-map based upon the informative tabulation done in Table 3 (page 15). In summary, what we have observed from our deep dissection of existing policies that any forthcoming development in this area needs to address all the shortcomings found in the current frameworks (as pointed out in Table 3), and software systems should notch up the level of user-experience through both interactive user involvement and customization, as well as a measured limitation on the flow of information.

Reviewer # 2, Concern # 5

Concern: Figure 1 is very unclear. Please replace it with a high-resolution picture.

Author Response: We updated the figure to a higher resolution. If we print out using colour printer then all the text are visible.

============

Thank You for the extensive time and effort that you have spent to provide us with such constructive feedback on range of issues. Each of your comments literally had highly commendable influence to make the work more detailed, insightful and complete. We extend our sincerest gratitude in acknowledging your critical contribution to our work.

____________________________________________

Reviewer 3 Report

The authors have analysed the existing privacy by design frameworks to identify the key limitations in healthcare sector. Seven contemporary privacy by design frameworks have been examined in-depth in this research based on systematic literature review. Though the article is detailed, but some key aspects of the literature review are missing. Therefore, article needs a major review, based on the following comments. 

  1. The "research problem" of the review paper is very short. It should explain in detail the security and privacy concerns plus how they are impacting the development of health care devices. The statement is too generic at the moment. Neither software vulnerabilities are mentioned nor the problems on the hardware level are enlisted. This is a major problem. For review article it should be very detailed. 
  2. The same stands true for "Aim" section. The author should introduce the existing frameworks in this section and should write the methodology of comparing all these frameworks in the later sections. 
  3. Figure 1 is totally un-necessary as it shows the publication count from specific databases this will not help the reader to evaluate the methodology of existing frameworks. It will be nice if authors enlist the frameworks in the table and mention key 5-6 references for each framework. 
  4. References are in-consistent. For example in line 156, authors  mention the contribution of Dr. Ann Cavoukian. It should be followed by the references such as the following. (a). Ann Cavoukian and Don Tapscott. 1996. Who Knows: Safeguarding Your Privacy in a Networked World. McGraw-Hill Professional. (b). A. Cavoukian, "Privacy by Design [Leading Edge]," in IEEE Technology and Society Magazine, vol. 31, no. 4, pp. 18-19, winter 2012.
  5. Line 168 mentioned the word "electronic data" is stored. It should be replaced with the word "digital". 
  6. The other important aspect that is missing is the industrial manufacturing and its impact on health sector. With the shift from industrial manufacturing to knowledge creation and service delivery, the value of information and the need to manage it responsibly have grown dramatically. At the same time, rapid innovation, global competition and increasing system complexity present profound challenges for informational privacy.
  7. Section 2.4.2 is too short. Again, the purpose of the review article is to give readers a broad spectrum of the topic. Privacy by Default, is particularly informed by the following FIPs (Fair Information Practices) that should be well explained in this section. For example, purpose specifications, collection limitations, data minimization, retention and disclosure agreement.
  8. The section 2.4.3 has the similar problem. More in depth discussion should be added around embedded system architecture and privacy my design issues.
  9. Section 2.5, the explanation for the design patterns are not enough especially for dynamic location granularity, encryption and k-anonymity.
  10. For section 2.6, consent, accuracy, authentication, access and compliance should be discussed.
  11. In the section 3.5, discussion is missing in-terms of IoT devices encryption, processing and guidelines for storage of data either at Edge or Fog level.
  12. The conclusion should be a comparative analysis of the frameworks rather than a generic importance of privacy in healthcare.
  13. The whole article needs proof-reading and no colons (:) should be placed after the headings.

Author Response

Reviewer # 3, Concern # 1

Concern: The "research problem" of the review paper is very short. It should explain in detail the security and privacy concerns plus how they are impacting the development of health care devices. The statement is too generic at the moment. Neither software vulnerabilities are mentioned nor are the problems on the hardware level enlisted. This is a major problem. For review article it should be very detailed?

Author Response: First of all thank you for providing a series of extremely detailed and helpful review comments.

We have made required modifications to the section in question. Your valued suggestion aided the research problem section to now become lot more dynamic and clearer.

Reviewer # 3, Concern # 2

Concern: The same stands true for "Aim" section. The author should introduce the existing frameworks in this section and should write the methodology of comparing all these frameworks in the later sections.

Author Response: Thank You for the opinion. We have thoroughly implemented the valuable advice.

Reviewer # 3, Concern # 3

Concern: Figure 1 is totally un-necessary as it shows the publication count from specific databases this will not help the reader to evaluate the methodology of existing frameworks. It will be nice if authors enlist the frameworks in the table and mention key 5-6 references for each framework.

Author Response: We highly appreciate your concern, however, due to the linkage and relation between Table 3 and Figure 1, the figure, we believe, makes the paper more complete.

Reviewer # 3, Concern # 4

Concern: References are in-consistent. For example in line 156, authors  mention the contribution of Dr. Ann Cavoukian. It should be followed by the references such as the following. (a). Ann Cavoukian and Don Tapscott. 1996. Who Knows: Safeguarding Your Privacy in a Networked World. McGraw-Hill Professional. (b). A. Cavoukian, "Privacy by Design [Leading Edge]," in IEEE Technology and Society Magazine, vol. 31, no. 4, pp. 18-19, winter 2012.

Author Response: The suggested references have been added and can be seen at number 105 and 106.

Reviewer # 3, Concern # 5

Concern: Line 168 mentioned the word "electronic data" is stored. It should be replaced with the word "digital".

Author Response: We updated the term, can be located at line # 207.

Reviewer # 3, Concern # 6

Concern: The other important aspect that is missing is the industrial manufacturing and its impact on health sector. With the shift from industrial manufacturing to knowledge creation and service delivery, the value of information and the need to manage it responsibly have grown dramatically. At the same time, rapid innovation, global competition and increasing system complexity present profound challenges for informational privacy.

Author Response: Thank You for one of the most impactful review comments. The manufacturing industry does play a substantial role as pointed out by you and consequently we have incorporated critical views on it, found under section 3.5.2, from line number 665 onward. This is an important insight into the paper that have indeed brought in a perspective from a different, but highly significant dimension.

Reviewer # 3, Concern # 7

Concern: Section 2.4.2 is too short. Again, the purpose of the review article is to give readers a broad spectrum of the topic. Privacy by Default, is particularly informed by the following FIPs (Fair Information Practices) that should be well explained in this section. For example, purpose specifications, collection limitations, data minimization, retention and disclosure agreement.

Author Response: The section in question indeed could have been explained and presented in a better and detailed fashion as you have rightly pointed out. Now we have implemented your advice which can be found at line number 262 onward.

Reviewer # 3, Concern # 8

Concern: The section 2.4.3 has the similar problem. More in depth discussion should be added around embedded system architecture and privacy my design issues.

Author Response: Thank you for the useful remark regarding the privacy issues in embedded systems. We have extended the relevant discussion on it from line number 277.

Reviewer # 3, Concern # 9

Concern: Section 2.5, the explanation for the design patterns are not enough especially for dynamic location granularity, encryption and k-anonymity.

Author Response: Necessary explanations have been added (line number 364, under section 2.5.1) to address the issue, these are some of the key components and inclusion of these clearly enriched the overall discussion.

Reviewer # 3, Concern # 10

Concern: For section 2.6, consent, accuracy, authentication, access and compliance should be discussed.

Author Response: Thank you for the comment, we have affected the advised changes (line number 463, under section 2.6.1).

Reviewer # 3, Concern # 11

Concern: In the section 3.5, discussion is missing in-terms of IoT devices encryption, processing and guidelines for storage of data either at Edge or Fog level.

Author Response: Yes we fully acknowledge the fact that in current times, privacy issues surrounding IoT devices are highly important, and thus we included a relevant section on it to address the concern (line number 665 onward, under section 3.5.2).

Reviewer # 3, Concern # 12

Concern: The conclusion should be a comparative analysis of the frameworks rather than a generic importance of privacy in healthcare.

Author Response: Conclusion has been updated to reflect the suggested changes, which now provides increased clarity regarding the distinctions and future trends.

Reviewer # 3, Concern # 13

Concern: The whole article needs proof-reading and no colons (:) should be placed after the headings.

Author Response: Colons have been removed and writing issues have been resolved after a thorough proof-reading.

============

Thank You for such in-depth and constructive feedback on such diverse range of issues. Each of your comments literally had highly commendable influence to make the work more detailed, insightful and complete. We extend our sincerest gratitude in acknowledging your critical contribution to our work.

____________________________________________

Round 2

Reviewer 1 Report

I think the authors have addressed my comments properly. I appreciate authors' efforts in finishing them in a short time. The paper has been greatly improved and can be published in this shape.

Reviewer 3 Report

The authors have addressed the questions in the revised manuscript. It will be a great read for the researchers. Therefore, I recommend to accept the paper.